# Factors influencing drug-susceptible tuberculosis treatment outcomes in Romania and Ukraine

Ioana Margineanu[1☯], Fajri Gafar[2,3,4,5☯], Teodora Butnaru[6], Dragos Baiceanu[6], Raluca Dragomir[6], Ihor Semianiv[7], Florin Mihaltan[6], Ioana Munteanu[6], Beatrice Mahler[6], Liliia Todoriko[7], Sorina Margineanu[8], Ymkje Stienstra[9,10], Jan-Willem C. Alffenaar[1,11,12,13], Onno W. Akkerman[14,15]*

**1** Department of Clinical Pharmacy and Pharmacology, University Medical Center Groningen, University of Groningen, Groningen, The Netherlands, **2** Unit of PharmacoTherapy, -Epidemiology and -Economics, Groningen Research Institute of Pharmacy, University of Groningen, Groningen, The Netherlands, **3** Respiratory Epidemiology and Clinical Research Unit, Centre for Outcomes Research and Evaluation, Research Institute of the McGill University Health Centre, Montreal, Quebec, Canada, **4** McGill International TB Centre, McGill University, Montreal, Quebec, Canada, **5** Department of Biomedical Sciences, Faculty of Medicine, Universitas Padjadjaran, Bandung, Indonesia, **6** Marius Nasta TB Institute, Bucharest, Romania, **7** Chernivtsi TB Expertise Centre, Bukovinian State Medical University, Chernivtsi, Ukraine, **8** Department of Computer and Data Sciences, "Ion Ionescu de la Brad" Iasi University of Life Sciences, Iasi, Romania, **9** Department of Internal Medicine/Infectious Diseases, University Medical Center Groningen, University of Groningen, Groningen, The Netherlands, **10** Department of Clinical Sciences, Liverpool School of Tropical Medicine, Liverpool, United Kingdom, **11** School of Pharmacy, Faculty of Medicine and Health, University of Sydney, Sydney, New South Wales, Australia, **12** Westmead Hospital, Westmead, New South Wales, Australia, **13** Sydney Institute for Infectious Diseases, University of Sydney, Sydney, New South Wales, Australia, **14** Department of Pulmonary Diseases and Tuberculosis, University Medical Center Groningen, University of Groningen, Groningen, The Netherlands, **15** Tuberculosis Center Beatrixoord, University Medical Center Groningen, University of Groningen, Haren, The Netherlands

☯ These authors contributed equally to this work.
* o.w.akkerman@umcg.nl

## Abstract

### Background

Tuberculosis (TB) remains one of the most globally impactful infectious diseases, with a recorded mortality of 1.6 million in 2022. In Romania and Ukraine, two high burden countries in the context of the WHO European region, treatment is geared towards cure; however, this path is paved with significant challenges, from morbidity to loss to follow-up.

### Methods

A retrospective study was performed for drug-susceptible TB patients hospitalised in three TB expertise centres in Romania and Ukraine using routinely collected data. Univariable and multivariable logistic regression analyses were used to assess

**Data availability statement:** All relevant data are within the paper and its Supporting Information files.

**Funding:** IM is funded through a doctoral project funded from the European Union Horizon 2020 research and innovation programme, under the Marie-Skłodowska Curie grant agreement 713660. The funding source had no impact on any decision-making regarding this paper.

**Competing interests:** The authors have declared that no competing interests exist.

predictors of three treatment outcomes: unfavourable outcomes, loss to follow-up, and death.

## Results

A total of 838 patients diagnosed with drug-susceptible TB were included. Median hospitalisation was 39 days (IQR 25–67), and treatment duration was 7 months (IQR 6–8). Predictive variables differed by outcome. For unfavourable outcomes, the multivariable model included age > 65 years, chronic kidney disease, at least one cavity on chest X-ray, underweight status, and persistently abnormal laboratory parameters despite intervention. Independent predictors of loss to follow-up were alcohol use, COPD, TB infection within two years prior to admission, obesity, slow treatment response, and sputum microscopy ≥2 + . Predictors of death included age > 65 years, male sex, cirrhosis, chronic kidney disease, underweight status, persistently abnormal laboratory parameters, and slow treatment response.

## Conclusion

Contextualising factors influencing drug-susceptible TB treatment outcomes in different settings can support the development of tailored interventions that enable early identification of patients at higher risk, thereby avoiding unnecessary treatment effects.

## Introduction

Tuberculosis (TB) remains one of the most globally impactful infectious diseases, with an estimated 1.6 million deaths in 2022 [1]. Ending TB starts with a better understanding of the risk factors for incidence, prevalence, morbidity, and mortality. A systematic review on studies involving prediction models reported that the most common predictors for poor outcomes in TB were age, sex, anatomical location of TB, body mass index, chest X-ray abnormalities, previous TB history, and HIV status [2]. The World Health Organization (WHO) recommends performing local research to tailor TB strategies for specific population needs, as health care settings, socio-economic conditions, and cultural settings differ [3].

The WHO TB European region has the fastest decline in TB incidence and mortality rates, and it is, overall, a low-incidence TB region. Yet, Romania and Ukraine still tackle a large number of TB cases, with Romania accounting for one-fourth of all TB cases in the EU [4] and Ukraine being in the top 30 WHO high TB burden countries [5]. The two countries have several commonalities, especially regarding the region of Bukovina, which was split between west Ukraine and north Romania after the Second World War. The shared geographical and cultural past ensures not only genetic commonalities between strains of *Mycobacterium tuberculosis* but also similarities in TB management approaches. Both countries have healthcare policies geared towards TB treatment success, including ensuring that patients are hospitalised until

sputum conversion and keeping in mind their socio-economic background and estimated capacity to follow through with treatment recommendations [6,7]. The success rates for drug-susceptible TB (DS-TB) are 80% in Romania and 75% in Ukraine, compared to 77% overall in the WHO European region [5,8,9]. This common path to treatment success is being paved with significant morbidity, hospital admissions for all TB-affected individuals, adverse events, and other unfavourable outcomes such as loss to follow-up (LTFU). Pre-pandemic TB-related mortality in Romania was 5 per 100,000 and in Ukraine 8.7 per 100,000, compared to the overall estimate for the WHO European region of 2.2 per 100,000 people [6]. Similarly, reported rates of LTFU were higher in Romania (4.6% between 2015 and 2016 [10]) and Ukraine (9.9% in one study performed in Kyiv between 2012 and 2014 [11]) compared to the WHO European region (3.9% in 2018) [12]. As no prediction model has been described for this setting [2], this study aims to explore factors associated with negative outcomes in patients treated for DS-TB in Romania and Ukraine.

## Methods

### Study design

A retrospective multi-center cohort study was conducted on adult patients (>18 years) diagnosed with any form of DS-TB, either bacteriologically confirmed by culture or diagnosed clinically, who initiated treatment between 1 January 2019 and 31 December 2019. This period was chosen as it was not affected by the COVID-19 pandemic or the war in Ukraine. To be eligible for inclusion, patients were required to have documented data, including routine laboratory tests (renal and hepatic values) performed at least twice (once at diagnosis and once during follow-up), and outcomes reported at the end of treatment. All consecutive eligible patients within the period specified were included.

### Settings

Three study sites participated in this study: two in Romania and one in Ukraine. These included the Bucharest Marius Nasta Institute and the Iasi Lung Hospital in Romania, and the Chernivtsi TB Centre in Ukraine. All three study sites are specialised TB clinics that receive patients from the wider surrounding regions. The study site in Ukraine was chosen as it offers a diverse patient population, whilst maintaining a higher chance of having *Mycobacterium tuberculosis* strains with genetic similarity to the Romanian ones.

The Marius Nasta Institute in Bucharest is the regional TB centre for the Muntenia region of southern Romania (population 2.9 million in 2019) [13] and serves as the main centre of TB expertise in the country. It provides care for both drug-sensitive and drug-resistant TB patients, primarily from the urban, higher-income region of Bucharest. The Iasi Lung Hospital is located in north-eastern Romania (population 3.19 million in 2019) [13]. Iasi is the second-largest city in Romania by population (500,668). The region has the lowest income in Romania, estimated to be 3.6 times lower than the Bucharest region and has a balanced rural-urban population (55.5% rural) [14]. Chernivtsi Hospital treats patients from the Bukovyna region (population 906,700 in 2018) [15] in southern Ukraine.

### Ethical considerations

The study was approved by the ethics committee of each centre (Iași: 5483/2021, Bucharest: 10592/2019, Chernivtsi: 234/2021). The need for written informed consent was waived due to the retrospective nature of this study.

### Data collection and storage

Data collection was performed between 5 June 2021 and 23 November 2021. Data included socio-demographics, smoking and alcohol use, comorbidities, previous TB history, TB diagnosis, hospitalisation data (including date of admission and discharge, laboratory parameters at treatment initiation and after 15–30 days of hospitalisation, and any side effects reported), and end-of-treatment outcomes. Patient data were coded and extracted from both digital and paper

medical records and reported as found in medical charts. Follow-up information and end-of-treatment outcomes post-hospitalisation were retrieved from national TB registries, using WHO standard definitions for treatment success (cured or completed treatment), treatment failure, LTFU, and death (see S1 Box) [16].

Data extraction was performed by a local investigator at each site. Subsequently, a second local investigator cross-verified the extracted data for accuracy and completeness. The final dataset was checked for inconsistencies. Data were collected using Research Electronic Data Capture (REDCap, version 11.0.3), a validated software widely used for clinical studies. Authors had access to identifying information only during data collection, as the data were entered into an anonymized REDCap form, and all analyses were performed on de-identified data (e.g., without names, social security numbers, or addresses).

### Data analysis

Data was analysed as a single dataset to ensure some measure of variability and generalisability for this region. Descriptive statistics were used to characterise the study population. Age cut-off values were decided based on previous research [17] and the most frequent definition of an elderly person [18]. Side effects reported were grouped per local criteria in the following categories: "mild allergic reaction", "severe allergic reaction", "hospital acquired infections", "neurological and psychiatric" and "gastrointestinal". Persistently abnormal laboratory parameters included any abnormal values present at 14 days after admission.

Univariable regression analyses were performed to determine the relationship of each independent variable with unfavourable outcomes (combined outcome of LTFU, treatment failure, and death) and LTFU. For mortality, Cox proportional-hazards regression models were used to analyze survival at the end of treatment and month of death. Multivariable logistic regression models were developed for each outcome using a stepwise approach. The first step involved backward deletion, starting with a base model including all candidate variables and sequentially removing collinear and least significant variables ($p < 0.5$). In the second step, variables were reintroduced ($p < 0.05$) until a final model was obtained in which all variables met the threshold of $p < 0.2$ [19,20]. Each iteration was followed by a goodness-of-fit assessment. We evaluated the goodness-of-fit of the final models using the Hosmer–Lemeshow test and assessed performance using the area under the receiver operating characteristic (ROC) curve. For Cox regression, proportional hazards assumptions were checked before fitting the model and using a log-minus-log survival curve after fitting the model. Additionally, Cox regression results are reported as adjusted hazard ratios (aHRs), while logistic regression results are presented as adjusted odds ratios (aORs) with corresponding 95% confidence intervals (95% CIs). All statistical analyses were conducted using SPSS Statistics (version 27).

## Results

### General characteristics

During the study period, a total of 838 patients diagnosed with DS-TB were included: 369 patients for both Iasi and Bucharest clinical sites, and 100 for Chernivtsi. Few patients (n = 21) were excluded due to missing hospitalisation data or were transferred out. Patients were predominantly adult males with 19.6% being over the age of 65 years. Most patients (n = 709, 84.6%) were newly diagnosed with TB. The majority of patients had pulmonary TB (n = 606, 72.3%), while chest X-ray findings mostly showed bilateral involvement (n = 480, 57.3%) (Table 1).
Homelessness was reported in 1.3% of patients, and 19.1% were underweight, defined as having a body mass index (BMI) of <18.5 kg/m². Half of the patients had at least one comorbidity at admission. The most frequently reported comorbidity was cardiovascular disease (26.6%), followed by COPD (12.6%) and gastrointestinal conditions (12.3%), with a minority living with HIV (2.3%). One-third of the patients (33.4%) were hospitalised for less than one month, with a median hospitalisation duration of 39 days (interquartile range [IQR] 25–67). Patients received treatment for a median of 7 months (IQR 6–8).

**Table 1. Demographics and clinical characteristics of drug-susceptible TB patients.**

| Characteristic | Number of patients with available data | Percentage, or median (IQR)* |
|---|---|---|
| **Age, years** | 838 | 47 (35-61)* |
| **Living area** | 838 | |
| Urban | 340 | 40.6% |
| Rural | 486 | 58% |
| Unhoused | 12 | 1.4% |
| **Sex** | 838 | |
| Male | 557 | 66.5% |
| Female | 281 | 33.5% |
| **Body mass index (BMI)** | 838 | 21 (19-22.9)* |
| **Nutritional status** | 838 | |
| Underweight, BMI < 18.5 | 160 | 19.1% |
| Normal weight, BMI 18.5–24.9 | 581 | 69.3% |
| Overweight, BMI 25.0–29.9 | 72 | 8.6% |
| Obese, BMI ≥ 30 | 25 | 3% |
| **Previous TB history** | 838 | |
| New case | 709 | 84.6% |
| Previous TB > 2 yrs ago | 99 | 11.8% |
| Previous TB < 2 yrs ago | 30 | 3.6% |
| **Smoking** | 761 | |
| Never | 283 | 33.8% |
| Former | 78 | 9.3% |
| Current | 400 | 47.7% |
| **Alcohol consumption** | 655 | |
| Never | 269 | 32.1% |
| Light drinking | 132 | 15.8% |
| Moderate drinking | 99 | 11.8% |
| Heavy drinking | 155 | 18.5% |
| **Clinical comorbidities** | 838 | |
| Cardio-vascular | 223 | 26.6% |
| COPD | 106 | 12.6% |
| Diabetes mellitus type 1 or 2 | 55 | 6.6% |
| Cancer | 31 | 3.7% |
| Cirrhosis | 23 | 2.7% |
| Chronic Kidney Disease | 18 | 2.1% |
| **Anatomical location of TB** | 838 | |
| Pulmonary | 606 | 72.3% |
| Pleura | 103 | 12.3 |
| Pulmonary + Pleura | 66 | 7.9% |
| Extrapulmonary | 27 | 3.2% |
| Miliary | 9 | 1.1% |
| Pulmonary + extrapulmonary | 27 | 3.2% |
| **TB pulmonary X-ray** | 698 | |
| Unilateral | 218 | 31.2% |
| Bilateral | 480 | 57.3% |

*(Continued)*

**Table 1.** (Continued)

| Characteristic | Number of patients with available data | Percentage, or median (IQR)* |
|---|---|---|
| **Direct microscopy of sputum** | 697 | |
| Positive | 499 | 59.5% |
| Negative | 198 | 23.6% |
| **Mycobacterial culture result** | 690 | |
| Positive | 639 | 92.6% |
| Negative | 51 | 6.1% |

COPD: chronic obstructive pulmonary disease; IQR: interquartile range; TB: tuberculosis.

## Treatment outcomes

All patients received standardised treatment regimens for DS-TB based on WHO guidelines [21]. Treatment success was achieved in 88.4% of patients (Table 2). In total, 11.3% experienced an unfavourable outcome. Treatment failure was reported in only 0.5% of cases, which precluded statistical analysis for this outcome. Additionally, 4.7% of patients were LTFU and 6.2% died.

## Univariable analysis

Several variables showed an association with one or more outcomes in the univariable analyses and qualified for inclusion in model development. These included age over 65 years, homelessness, TB retreatment, the presence of liver cirrhosis or chronic kidney disease, being underweight (BMI < 18.5 kg/m$^2$) or obese (BMI > 30 kg/m$^2$), a sputum microscopy result of more than 9 bacilli per field (+++), gastrointestinal side effects during hospitalisation (e.g., nausea), and abnormal liver enzyme or renal function parameters. Detailed univariable analysis results for unfavourable outcomes, LTFU, and death are presented in S1–S3 Tables.

## Development of multivariable models

The initial model for unfavourable outcomes included 23 variables. Variables were removed based on significance and by eliminating collinear variables, such as alcohol consumption, liver cirrhosis, and elevated liver enzymes at diagnosis, or obesity and type 2 diabetes mellitus. After backward deletion, a set of four variables remained, and following the reintroduction of variables, the final model included five. A total of 64 iterations were performed to obtain this final model. For LTFU, the initial model included 25 variables, which were reduced to four after backward deletion. Following reintroduction and a total of 84 iterations, the final model included five variables. The model for survival until the end of treatment initially included 17 variables, which were reduced to six after backward deletion. After 55 iterations, the final model with seven variables was obtained.

## Predictors included in multivariable models

The final multivariable model for unfavourable outcomes (goodness-of-fit: Hosmer–Lemeshow test p = 0.934; area under the ROC curve [AUC] = 0.784, 95% CI: 0.700–0.869) included five variables: age over 65 years, chronic kidney disease as a comorbidity, presence of at least one cavity on chest X-ray, underweight status, and persistently abnormal laboratory parameters despite hospital interventions (Table 3). Variables with higher p values (p > 0.05) were retained due to the model-building algorithm and their notable effect sizes.

**Table 2. Hospitalisation and treatment outcomes.**

| Characteristic | Number of patients included | Percentage, or median (IQR)* |
|---|---|---|
| **Hospitalisation time, days** | 838 | 39 (25-67)* |
| Iasi | 369 | 34 (18-64) |
| Bucharest | 369 | 37 (27-57) |
| Chernivtsi | 100 | 92.5 (69-105) |
| **Laboratory values** | 838 | |
| Abnormal laboratory values at baseline¶ | 223 | 26.6% |
| Abnormal laboratory values during hospitalisation | 321 | 38.3% |
| **Follow-up of laboratory parameters** | 248 | |
| Abnormal values returned to baseline | 174 | 70% |
| Values improved, but never reached baseline | 47 | 19% |
| Persistently abnormal laboratory values | 27 | 10.9% |
| **Clinical management of abnormal values** | 221 | |
| No attitude/expectancy | 33 | 14.9% |
| Supportive treatment§ | 83 | 37.6% |
| Lowered dosages of TB medication | 30 | 13.6% |
| Paused TB drugs <= 7 days | 20 | 9.0% |
| Paused TB drugs > 7 days | 31 | 14% |
| Stopped/switched TB medication | 24 | 16.7% |
| Slow response to treatment | 192 | 22.9% |
| **End-of-treatment outcomes** | 838 | |
| Treatment success | 741 | 88.4% |
| Loss to follow-up | 39 | 4.7% |
| Death | 52 | 6.2% |
| Treatment failure | 4 | 0.5% |
| Transfer out | 2 | 0.2% |

¶Listed if at least one parameter of ALAT, ASAT, eGFR (CKD-EPI) was out of normal range.

§Supportive treatment includes hydration, iv infusions including thiamine, folic acid, magnesium, glucose, hepatoprotectors.

**Table 3. Multivariable logistic regression model for unfavourable outcomes*.**

| Characteristic | aOR | 95% CI | p-value |
|---|---|---|---|
| Age > 65 years | 2.02 | 0.76−5.97 | 0.157 |
| Chronic kidney disease | 11.53 | 2.43−54.66 | 0.002 |
| Cavity present on chest X-ray | 1.78 | 0.74−4.30 | 0.196 |
| BMI < 18.5 kg/m$^2$ (underweight) | 5.59 | 2.29−13.67 | <0.001 |
| Persistently abnormal laboratory tests | 6.30 | 2.13−18.65 | 0.001 |

aOR: adjusted odds ratio, BMI: body mass index, CI: confidence interval.

* Goodness-of-fit Hosmer−Lemeshow test: p = 0.934. Area under the receiver operating characteristic (ROC) curve: 0.784, 95% CI 0.700−0.869.

For predictors of LTFU, the final multivariable model (goodness-of-fit: Hosmer–Lemeshow test p = 0.871; AUC = 0.860, 95% CI: 0.800–0.921) included six variables: alcohol consumption, COPD as a comorbidity, TB disease history within two years prior to admission, obesity, slow treatment response, and a sputum microscopy result of at least 2+ (Table 4).

**Table 4. Multivariable logistic regression model for loss to follow-up[*].**

| Characteristic | aOR | 95% CI | p-value |
|---|---|---|---|
| Alcohol consumption | 4.99 | 1.41–17.02 | 0.012 |
| COPD | 3.70 | 1.26–10.86 | 0.017 |
| Previous TB < 2 years ago (rapid relapse) | 14.48 | 3.37–56.11 | <0.001 |
| BMI > 30 kg/m$^2$ (obesity) | 17.16 | 3.84–76.66 | <0.001 |
| Culture conversion at month two | 13.21 | 2.37–73.57 | 0.003 |
| Ziehl-Nielsen sputum result of at least 2+ | 4.45 | 1.37–14.48 | 0.013 |

aOR: adjusted odds ratio, BMI: body mass index, COPD: chronic obstructive pulmonary disease, CI: confidence interval.

[*]Goodness-of-fit Hosmer−Lemeshow test: p = 0.871. Area under the receiver operating characteristic (ROC) curve: 0.860, 95% CI 0.800−0.921.

For predictors of death, the final multivariable model included seven variables: age over 65 years, male sex, cirrhosis or chronic kidney disease as comorbidities, underweight status, persistently abnormal laboratory parameters, and slow treatment response (Table 5).

## Discussion

In this study, retrospective data from TB patients at three centres in Eastern Europe were analysed to develop multivariable models for the composite outcome of unfavourable treatment outcomes and two of its components: death and LTFU. Variables included in the unfavourable outcomes model were age over 65 years, chronic kidney disease, the presence of at least one cavity on chest X-ray, underweight status, and persistently abnormal laboratory parameters despite hospital interventions. Four of these variables were also present in the death model, while none overlapped with the LTFU model.

Overall treatment success rate in our DS-TB population was higher than the reported national indicators for 2019. In our study, treatment success was achieved in 88.4% of patients, compared with 79.8% in Romania, 75.2% in Ukraine, and 76.5% in the broader WHO European Region [5,22]. The higher success rate observed in our cohort may be partly explained by the inclusion of patients with DS-TB treated in specialised TB centres. Similar findings have been reported in other studies from Ethiopia and South Africa, where TB patients tended to achieve better outcomes in specialised or private clinics [23,24].

Treatment success in our study was achieved following hospitalization of all TB patients for a median duration of 39 days during the intensive phase and nearly eight months of total treatment duration. In both Romania and Ukraine, the standard practice involves hospitalisation until culture conversion and/or until there is a reasonable assurance that patients can continue treatment at home [6,7]. Another study in Ukraine evaluating extended therapy for DS-TB reported

**Table 5. Multivariable Cox regression model for mortality.**

| Characteristic | aHR | 95% CI | p-value |
|---|---|---|---|
| Age > 65 years | 5.73 | 3.00–10.92 | <0.001 |
| Male | 2.22 | 1.08–4.56 | 0.030 |
| Cirrhosis | 3.54 | 1.27–9.88 | 0.016 |
| Chronic kidney disease | 7.65 | 2.75–21.52 | <0.001 |
| BMI < 18.5 kg/m$^2$ (underweight) | 7.66 | 3.89–15.07 | <0.001 |
| Persistent abnormal lab values | 7.28 | 2.60–20.38 | <0.001 |
| Slow responders | 7.18 | 3.61–14.28 | <0.001 |

aHR: adjusted hazard ratio, BMI: body mass index, CI: confidence interval.

a 100% success rate among patients with negative cultures at month two, after a median treatment duration of 275 days (nine months), whereas those who remained culture-positive at month two had a 74.5% success rate [11].

Global recommendations, however, are shifting toward shorter treatment regimens, as evidence suggests these may achieve comparable success rates while minimising treatment burden [21]. In contrast, prolonged treatment has been associated with increased risks of drug toxicity and LTFU [25]. In our cohort, 15.5% of patients experienced at least one adverse event during hospitalisation, ranging from mild allergic reactions to hospital-acquired infections. However, despite being associated in the univariable analysis with negative outcomes, gastro-intestinal side effects were not associated in the multivariable models.

Our model for unfavourable outcomes is comparable to earlier models [2], which identified age, sex, comorbidities, and being underweight or malnourished as significant predictors. However, our population had several distinctive features. For instance, unlike most studies conducted in high TB-HIV burden settings, our cohort represented a TB population with a low HIV prevalence (2.3%). As a result, HIV infection and CD4 count did not contribute meaningfully to the model predicting unfavourable outcomes, in contrast to findings from other populations [26–29].

In the EU/EEA, previous models have identified predictors for unfavourable outcomes such as male sex, older age, alcohol abuse, a history of mental disorders, bilateral lung involvement, anaemia, and the presence of at least one significant comorbidity (e.g., in Denmark [30] and Portugal [26]). In the broader WHO European region, a retrospective study conducted in Russia between 1993 and 2002 identified sex, unemployment, retreatment status, alcohol abuse, severe TB forms, place of residence, age, pulmonary TB (vs. extrapulmonary), and a history of imprisonment as predictors of poor outcomes [31]. In our study, the composite endpoint of unfavourable outcomes included a higher proportion of deaths than LTFU, which may explain the greater overlap of variables between the models for unfavourable outcomes and death, and the fewer shared predictors with LTFU. As in the Russian study, retreatment was included in our LTFU model, but only when the previous TB episode had occurred within the past two years, possibly reflecting the burdensome nature of TB treatment. At the same time, predictors might have changed in Russia as well, as the study mentioned was performed 10 years prior.

Concerning LTFU, in our study, alcohol consumption, COPD, rapid relapse, obesity, and a sputum smear result of at least 2+ were associated with an increased risk of LTFU. A study conducted in Georgia [32] reported several overlapping factors but differed by identifying unemployment and male sex as predictors, whereas in our analysis, male sex was associated with mortality. In a 5-year observational study performed in China [33], factors associated with LTFU were negative AFB smear/GeneXpert/culture results, chronic hepatitis or cirrhosis, the occurrence of adverse drug reactions and being employed/self-employed. Interestingly, in our study, timely culture conversion before the end of the intensive phase was associated with an increased risk of LTFU, a finding not identified by other studies. This suggests that patients who experience rapid clinical improvement might perceive themselves as "cured", leading to premature discontinuation of treatment. Other studies from non-European settings have reported that the risk of LTFU increases among older individuals with low income and a history of previous treatment interruption (e.g., Iraq [34]), or among younger males receiving care at primary health clinics in border or transit regions (e.g., Namibia [35]). In our study, however, age was not retained in the final multivariable model for LTFU, as it was not statistically significant in either the univariable or multivariable analyses.

Concerning predictors of mortality, our model included older age, being male, having cirrhosis or chronic kidney disease at diagnosis, being underweight, being a slow responder, and having persistently abnormal laboratory values. Regarding the latter, in 30% of included patients with abnormal liver enzymes, laboratory parameters did not normalise during hospitalisation. These data, typically collected between days 14 and 30 of hospital stay, help assess hospitalisation effects and drug toxicity. In Romania and Ukraine, strategies to mitigate drug toxicity include supportive treatment and modification of the TB regimen by pausing or removing drugs. However, when laboratory parameters do not improve despite these interventions, our models indicated that persistent abnormality of laboratory parameters during treatment was a predictor associated with unfavourable outcomes, including death. In such cases, therapeutic drug monitoring, which provides

information on drug plasma concentrations, may be particularly useful, as it can more precisely guide treatment decisions [36].

The model for mortality in our study is comparable to other studies, with other models developed in different populations showing overlap in certain predictors, such as age or comorbidities, but also differences, such as having anaemia or being homeless [37]. Variations in predictors may reflect the specific characteristics of our study centres, which are located in middle-income countries with high TB burdens. These findings underscore the importance of conducting localised research to inform tailored clinical recommendations.

A key strength of this study is its large cohort, which is distinctive for representing a high TB burden within the European region and for being drawn from resource-limited settings. The similarity in TB cases and TB treatment approaches allows for a certain measure of generalisability within the geographic and socio-economic context of Eastern Europe. The data analysed were routinely collected across all participating centres, making it feasible to inform clinicians about patients at increased risk. The stepwise approach used for model development is among the most widely applied in medical research, although it has inherent caveats. Additionally, the study captured information relevant to healthcare policy, including hospitalisation and treatment duration for TB patients in these countries.

This study also has several limitations that should be acknowledged. It relied on data entered by clinicians into medical charts and the national TB register. To mitigate limitations associated with this approach, we included only patients with documented follow-up and ensured consistency in data collection through the use of REDCap electronic forms, as none of the clinical centres had a comprehensive electronic medical record system. Although this remains a potential limitation, only 21 patients were excluded due to missing data. Furthermore, certain patient characteristics, such as socio-economic and educational background, were not included in medical charts. As the data collected were limited to routinely available information found in medical charts, no additional causality analysis was performed between adverse events and TB treatment, and they were reported descriptively. Additionally, selection bias could have been introduced due to differences in the number of patients included per study centre. However, when dividing the number of included patients by the population served by each study site, the results were 0.00011 for Ukraine, 0.0096 for Iasi and 0.00012 for Bucharest.

## Conclusion

In conclusion, our study identified distinct combinations of routinely collected variables that best predict unfavourable outcomes, death, and LTFU among TB patients in Romania and Ukraine. The three models highlight related factors that may contribute to specific outcomes and warrant careful clinical consideration. These findings can support clinicians in identifying patients at increased risk of poor outcomes. The differences in predictive variables for unfavourable outcomes, LTFU, and death underscore the importance of tailoring interventions to the specific needs of the target population in order to improve treatment outcomes.

## Supporting information

**S1 Box. Outcome definitions.**
(PDF)

**S1 Table. Univariate logistic regression analyses for unfavourable outcomes.**
(PDF)

**S2 Table. Univariate logistic regression analyses for loss to follow-up (LTFU).**
(PDF)

**S3 Table. Univariate Cox regression analyses for death.**
(PDF)

**S1 File. Anonymized dataset used in the analysis.**

(XLS)

## Acknowledgments

We appreciate the support of the REDCap research information management team at the University Medical Centre Groningen for their support with the REDCap mobile app.

## Author contributions

**Conceptualization:** Ioana Margineanu, Jan-Willem C. Alffenaar, O.W. Akkerman.

**Data curation:** Ioana Margineanu, Teodora Butnaru, Dragos Baiceanu, Raluca Dragomir, Ihor Semianiv, Sorina Margineanu.

**Formal analysis:** Ioana Margineanu, Fajri Gafar.

**Funding acquisition:** Ioana Margineanu.

**Investigation:** Ioana Margineanu.

**Methodology:** Ioana Margineanu, Fajri Gafar, Ymkje Stienstra, Jan-Willem C. Alffenaar, O.W. Akkerman.

**Project administration:** Ioana Margineanu, Florin Mihaltan, Ioana Munteanu, Beatrice Mahler, Liliia Todoriko.

**Supervision:** Fajri Gafar, Ymkje Stienstra, Jan-Willem C. Alffenaar, O.W. Akkerman.

**Writing – original draft:** Ioana Margineanu.

**Writing – review & editing:** Ioana Margineanu, Fajri Gafar, Teodora Butnaru, Dragos Baiceanu, Raluca Dragomir, Ihor Semianiv, Florin Mihaltan, Ioana Munteanu, Beatrice Mahler, Liliia Todoriko, Sorina Margineanu, Ymkje Stienstra, Jan-Willem C. Alffenaar, O.W. Akkerman.

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
