## [Decision Letter · Decision Letter 0]

8 Sep 2025

Dear Dr. Akkerman,

Thank you for submitting your manuscript to PLOS ONE. After careful consideration, we feel that it has merit but does not fully meet PLOS ONE’s publication criteria as it currently stands. Therefore, we invite you to submit a revised version of the manuscript that addresses the points raised during the review process.

Please submit your revised manuscript by Oct 23 2025 11:59PM. If you will need significantly more time to complete your revisions, please reply to this message or contact the journal office at plosone@plos.org . A rebuttal letter that responds to each point raised by the academic editor and reviewer(s). You should upload this letter as a separate file labeled 'Response to Reviewers'.A marked-up copy of your manuscript that highlights changes made to the original version. You should upload this as a separate file labeled 'Revised Manuscript with Track Changes'.An unmarked version of your revised paper without tracked changes. You should upload this as a separate file labeled 'Manuscript'.

We look forward to receiving your revised manuscript.

Kind regards,

Frederick Quinn

Academic Editor

PLOS ONE

2. In the online submission form, you indicated that [Data cannot be shared publicly because of ethical restriction. Investigators wishing to access deidentified patient data analyzed in this study will need to have a research proposal approved by their ethics committee and complete a data sharing agreement. All inquiries should be sent to the corresponding author (OA; o.w.akkerman@umcg.nl) and first author (IM; ismargineanu@gmail.com) upon written request.].

Additional Editor Comments (if provided):

Reviewer #1:

Reviewer #2:

Reviewers' comments:

Reviewer's Responses to Questions

**Comments to the Author**

1. Is the manuscript technically sound, and do the data support the conclusions?

Reviewer #1: Yes

Reviewer #2: Partly

2. Has the statistical analysis been performed appropriately and rigorously?

Reviewer #1: Yes

Reviewer #2: Yes

3. Have the authors made all data underlying the findings in their manuscript fully available?

Reviewer #1: No

Reviewer #2: No

4. Is the manuscript presented in an intelligible fashion and written in standard English?

Reviewer #1: Yes

Reviewer #2: Yes

Reviewer #1: This manuscript presents a valuable and well-conducted retrospective multi-center study analyzing predictors of unfavorable treatment outcomes in drug-susceptible tuberculosis (DS-TB) patients in two high-burden countries within the WHO European region: Romania and Ukraine. The topic is highly relevant, as identifying local predictors is crucial for tailoring interventions and improving TB control programs. The study is methodologically sound, with a clear design, robust statistical analysis, and a large, well-characterized cohort. The findings are novel for this specific regional context and have practical implications for clinicians and policymakers. I recommend acceptance after minor revisions to address the points outlined

Reviewer #2: Comments for the Author's Consideration

The manuscript "Factors influencing drug-susceptible tuberculosis outcomes in Romania and Ukraine” aimed to explore factors associated with negative outcomes in patients treated for DS-TB in Romania and Ukraine, and describes a retrospective study which was performed for drug-susceptible TB patients hospitalised in three TB expertise centers between 1 January 2019 to 31 December 2019. The manuscript is well-written, and data analysis appears to be carefully prepared, however, there are several shortcomings.

What new this study adds? Unfortunately, not much. TB treatment failure risk factors such as advanced age, sex, anatomical location of TB, advanced disease, body mass index, chest X-ray abnormalities, previous TB history, and HIV and diabetes status have been reported in numerous studies across the world. While the Authors correctly pointed out that “The World Health Organization (WHO) recommends performing local research to tailor TB strategies for specific population needs, as health care settings, socio-economic conditions, and cultural settings differ”, the study simply lumped together data from two different countries with different reported treatment success rates for DS-TB (80% in Romania and 75% in Ukraine). Romania is a member of EU, but Ukraine is not, however, no information about possible differences in the healthcare system or TB management is provided. Moreover, regarding Romania, the Authors mentioned that Iasi region has the lowest income in Romania, estimated to be 3.6 times lower than the Bucharest region; however, again, patient data from both regions/medical centers were lumped together.

No information on the M. tuberculosis drug susceptibility testing is provided.

No information on TB treatment in studied patients is provided (i.e. medications, dosage, regimes).

It is unclear how the majority of the reported risk factors (i.e. age, sex, BMI, slow response to treatment) are associated with “healthcare or cultural settings”.

Also, there is an apparent overlapping in study settings and time period with the published study by Margineanu et al. 2023 (doi: 10.5588/ijtld.22.0667). It was previously reported that “Between 1 January 2019 and 31 December 2020, there were 927 TB patients admitted in all three centres.” (i.e. the Iasi Lung Hospital (Iasi, Romania), Bucharest Marius Nasta Institute (Bucharest, Romania) and Chernivtsi TB Centre (Chernivtsi, Ukraine).). However, the present study reports that “During the study period, a total of 838 patients diagnosed with DS-TB were included”. Thus, it raises question, what was the total number of adult patients (>18 years) diagnosed with TB (including DS-TB) and initiated treatment between 1 January 2019 to 31 December 2019 in each of three medical centers, and raises concern either on data overlapping between the studies or patient selection process.

Minor comments:

In the Abstract, global TB mortality data for year 2020 are provided, which is, probably, a bit outdated. In contrary, in Introduction, data for year 2022 are provided, suggesting of a slightly careless attitude towards manuscript preparation and revision.

The timeframe of the study must be outlined in the Abstract because of the recent significant changes in geopolitical situation leading to the changes in TB management.

Please indicate how many patients experienced slow response to treatment, how many lived in rural areas, how many had diabetes, how many were obese or underweight. Please indicate median hospitalisation duration time for each medical center.

Probably, Authors would wish to revise Authorship credits. Seven supervisors - it sounds a bit excessive, but of course I may be wrong.

**Do you want your identity to be public for this peer review?** For information about this choice, including consent withdrawal, please see our Privacy Policy

Reviewer #1: **Yes: ** Richard Delali Agbeko Djochie

Reviewer #2: No

---

## [Author Response · Author response to Decision Letter 1]

22 Oct 2025

Note: All page, line, and reference numbers indicated in the response sections of this letter refer to the clean version of the revised manuscript without track changes.

Journal requirements (Editor):

Response: We have prepared the manuscript according to PLOS ONE style requirements.

2. In the online submission form, you indicated that [Data cannot be shared publicly because of ethical restriction. Investigators wishing to access deidentified patient data analyzed in this study will need to have a research proposal approved by their ethics committee and complete a data sharing agreement. All inquiries should be sent to the corresponding author (OA; o.w.akkerman@umcg.nl) and first author (IM; ismargineanu@gmail.com) upon written request.].

All PLOS journals now require all data underlying the findings described in their manuscript to be freely available to other researchers, either: 1. In a public repository, 2. Within the manuscript itself, or 3. Uploaded as supplementary information.

Response: We have now included the anonymized dataset as a supplementary file.

Response: The reviewer comments does not include a recommendation to cite specific previously published works.

REVIEWER #1:

Overall Assessment:

This manuscript presents a valuable and well-conducted retrospective multi-center study analyzing predictors of unfavorable treatment outcomes in drug-susceptible tuberculosis (DS-TB) patients in two high-burden countries within the WHO European region: Romania and Ukraine. The topic is highly relevant, as identifying local predictors is crucial for tailoring interventions and improving TB control programs. The study is methodologically sound, with a clear design, robust statistical analysis, and a large, well-characterized cohort. The findings are novel for this specific regional context and have practical implications for clinicians and policymakers. I recommend acceptance after minor revisions to address the points outlined below.

Response: Thank you for your very kind assessment.

Abstract:

1) Authors use the 2020 global TB mortality data in the abstract but 2022 data in the introduction. It is recommended that authors be consistent, preferably using the most recent (2022) data in both sections for accuracy.

Response: Thank you for noticing this. It has been addressed (line 46 of the Abstract section and line 71 of the Introduction section)

Introduction:

1) The introduction establishes the global and regional context but would be strengthened by a more specific rationale for focusing on Romania and Ukraine. Please add a sentence stating the specific LTFU and death rates for DS-TB in these two countries (citing national reports or WHO data) to quantitatively establish the magnitude of the problem and the rationale for this study.

Response: Thank you for your suggestion. We have added text in the introduction section reporting on the best evidence on mortality and LTFU in Romania and Ukraine (lines 93-97), as follows: “Pre-pandemic TB-related mortality in Romania was 5 per 100,000 and in Ukraine 8.7 per 100,000, compared to the overall estimate for the WHO European region of 2.2 per 100,000 people [6]. Similarly, reported rates of LTFU were higher in Romania (4.6% between 2015 and 2016 [10]) and Ukraine (9.9% in one study performed in Kyiv between 2012 and 2014 [11]) compared to the WHO European region (3.9% in 2018) [12].”

Methods:

1) Line 99: "follow up" should be "follow-up".

Reply: Thank you, we have made the correction (now line 93).

2) Please provide the rationale for selecting two centers in Romania and one in Ukraine. Furthermore, please state the proportion of patients contributed by each center in the Methods section (this data is in the Results, line 153-154) and discuss if any potential selection bias could arise from this uneven distribution.

Reply: Thank you for this comment. We have incorrectly initially reported the area served by the Marius Nasta centre and have corrected it and adjusted the population of all centres, and provided references. It has been addressed now, and the details are as follows:

In lines 113-118 of the Methods section: “Three study sites participated in this study: two in Romania and one in Ukraine. These included the Bucharest Marius Nasta Institute and the Iasi Lung Hospital in Romania, and the Chernivtsi TB Centre in Ukraine. All three study sites are specialised TB clinics that receive patients from the wider surrounding regions. The study site in Ukraine was chosen as it offers a diverse patient population, whilst maintaining a higher chance of having Mycobacterium tuberculosis strains with genetic similarity to the Romanian ones.”;

in lines 180-181 of the results section: “During the study period, a total of 838 patients diagnosed with DS-TB were included: 369 cases for both Iasi and Bucharest clinical sites, and 100 for Chernivtsi.”

in lines 337-339 of the discussion section: “The similarity in TB cases and TB treatment approaches allows for a certain measure of generalisability within the geographic and socio-economic context of Eastern Europe.”

and in lines 355-358 of the discussion section: “Additionally, selection bias could have been introduced due to differences in the number of patients included per study centre. However, when dividing the number of included patients by the population served by each study site, the results were 0.00011 for Ukraine, 0.0096 for Iasi and 0.00012 for Bucharest.”

3) Please clarify the sampling methodology. The manuscript states that patients "initiated treatment between 1 January 2019 to 31 December 2019." Does this imply that all consecutive eligible patients diagnosed within this one-year period were included? If not, what specific sampling or exclusion criteria were applied?

Response: We have added the clarification in the methods section, as follows (lines 103-110): “A retrospective multi-center cohort study was conducted on adult patients (>18 years) diagnosed with any form of DS-TB, either bacteriologically confirmed by culture or diagnosed clinically, who initiated treatment between 1 January 2019 and 31 December 2019. This period was chosen as it was not affected by the COVID-19 pandemic or the war in Ukraine. To be eligible for inclusion, patients were required to have documented data, including routine laboratory tests (renal and hepatic values) performed at least twice (once at diagnosis and once during follow-up), and outcomes reported at the end of treatment. All consecutive eligible patients within the period specified were included.

4) Lines 109 – 112: Kindly cite the references for the population statistics quoted for the study sites.

Response: We have added the references for study sites, as follows (lines 120-128): “The Marius Nasta Institute in Bucharest is the regional TB centre for the Muntenia region of southern Romania (population 2.9 million in 2019) [13] and serves as the main centre of TB expertise in the country. It provides care for both drug-sensitive and drug-resistant TB patients, primarily from the urban, higher-income region of Bucharest. The Iasi Lung Hospital is located in north-eastern Romania (population 3.19 million in 2019) [13]. Iasi is the second-largest city in Romania by population (500,668). The region has the lowest income in Romania, estimated to be 3.6 times lower than the Bucharest region and has a balanced rural-urban population (55.5% rural) [14]. Chernivtsi Hospital treats patients from the Bukovyna region (population 906,700 in 2018) [15] in southern Ukraine.”

5) Lines 112 – 114: The reference for the statement that the region where Iasi is located has the lowest income in Romania should be cited.

Response: Se our response to question 4, which has been addressed in the revised manuscript.

6) Line 115: "Data collection" header is misaligned.

Response: This has been addressed, thank you (now line 134)

7) What data collection tool was used; was it adopted from literature or was it self-developed by the authors? If it was self-developed, was it pre-tested or validated for this study?

Response: We have clarified the software used for data collection in lines 147-148, as follows: “Data were collected using Research Electronic Data Capture (REDCap, version 11.0.3), a validated software widely used for clinical studies.”

8) Line 118: Authors mentioned that they collected “hospitalization data”. They should kindly explain what specific data this category encompassed and how it differed from the other data types mentioned (e.g., socio-demographics, lab parameters).

Response: We included lab parameters in hospitalisation data, but also clarified what additional data we collected in line 135-143: “Data included socio-demographics, smoking and alcohol use, comorbidities, previous TB history, TB diagnosis, hospitalisation data (including date of admission and discharge, laboratory parameters at treatment initiation and after 15–30 days of hospitalisation, and any side effects reported), and end-of-treatment outcomes. Patient data were coded and extracted from both digital and paper medical records and reported as found in medical charts. Follow-up information and end-of-treatment outcomes post-hospitalisation were retrieved from national TB registries, using WHO standard definitions for treatment success (cured or completed treatment), treatment failure, LTFU, and death [16].”

9) Lines 125-126: For clarity, please rephrase the data verification process. For example: "Data extraction was performed by a local investigator at each site. Subsequently, a second local investigator cross-verified the extracted data for accuracy and completeness."

Response: Thank you for this advice, we have included it in lines 145-147 of the revised manuscript: “Data extraction was performed by a local investigator at each site. Subsequently, a second local investigator cross-verified the extracted data for accuracy and completeness. The final dataset was checked for inconsistencies.”

10) Line 126: The initials of the team member who cross-checked the data (IM) should be deleted to maintain anonymity.

Response: This has been addressed, and the initials have been removed - thank you.

11) Lines 141 – 142: Inconsistent use of the symbol p and P. Authors should stick to the standard symbol of lowercase 'p' throughout the text.

Response: Thank you for this suggestion. We have now used the standard symbol of lower case for p value throughout the manuscript.

12) Line 133 – 134: What is the reference for the age cut-off points used?

Response: This has been addressed in lines 155-157 of the data analysis, as follows: “Age cut-off values were decided based on previous research [17] and the most frequent definition of an elderly person [18].”

Results:

1) Line 158-159: "extrapulmonary TB cases involved the pleura (75.8%), either alone or in combination with pulmonary TB" – This is slightly confusing. It would be clearer to state the proportion of all patients who had pleural involvement.

Response: It has been addressed, and the details are now described in Table 1.

2) Lines 183 – 184: How did authors ascertain the causality of adverse drug reactions to anti-TB drugs? What criteria (e.g., WHO-UMC, Naranjo) were used in determining causality and severity? Please provide the authors’ specific definition of “abnormal liver enzymes and renal function” and “persistently abnormal lab parameters” (e.g., values outside normal range for a specific duration).

Reply: Causality was not investigated in this study, as it is not routinely reported in patient charts. We have clarified the definitions for side effects in lines 157-160 of the data analysis, as follows: “Side effects reported were grouped per local criteria in the following categories: “mild allergic reaction”, “severe allergic reaction”, “hospital acquired infections”, “neurological and psychiatric” and “gastrointestinal”. Persistently abnormal laboratory parameters included any abnormal values present at 14 days after admission.”

We have also added text in the limitation section, as follows (lines 353-355): “As the data collected were limited to routinely available information found in medical charts, no additional causality analysis was performed between adverse events and TB treatment, and they were reported descriptively.”

3) Table 2: The header "Percentage, or median (IQR)*" is used, but the "Laboratory values" section lists counts and percentages, not medians. The table formatting could be adjusted for clarity.

Response: For clarification, the second column lists the number of patients included, and the third column lists percentage or median values. Regarding “laboratory values”, we present the number and percentages of patients with abnormal laboratory values at baseline and during hospitalization.

4) The results for multivariable models are presented clearly. However, for variables like "Age >65 years" and "Cavity present" in Table 3, which have p-values > 0.05 (0.157 and 0.196), it is important to reiterate in the text that they were retained due to the pre-specified model-building algorithm (P<0.2), even though they are not "statistically significant" at the conventional 0.05 level. Their effect sizes (aORs) are still notable.

Response: Thank you for this very kind comment, we have adjusted the text accordingly, lines 229-230, as follows: “Variables with higher p values (p>0.05) were retained due to the model-building algorithm and their notable effect sizes.”

5) Line 210: "hystory" should be "history".

Response: This has been addressed, thank you.

Discussion:

1) Lines 232-234: The authors state the high success rate was achieved after hospitalizing all patients for a median of 39 days. This suggests a different standard of care. Please clarify in the Methods or Discussion: a) Is this hospitalization a mandatory national policy in Romania and Ukraine? b) What is the rationale for this approach? c) Does the median 39 days represent the entire treatment or just the initial intensive phase?

Response: Thank you for this comment. We have clarified in lines 261-265: “Treatment success in our study was achieved following hospitalization of all TB patients for a median duration of 39 days during the intensive phase and nearly eight months of total treatment duration. In both Romania and Ukraine, the standard practice involves hospitalisation until culture conversion and/or until there is a reasonable assurance that patients can continue treatment at home [6,7].”

2) The discussion is somewhat fragmented and would benefit from a more structured comparison with existing literature. For each key finding (e.g., success rate, predictors like obesity/LTFU, culture conversion/LTFU), please first provide a direct comparison with data from similar studies, then offer plausible reasons for any agreement or divergence. The connection to similar studies (e.g., the reference to Russia) could be expanded.

Response: Thank you for your comment. We have expanded and restructured the discussion section by adding each key finding in separate paragraphs, and in comparison, with data from similar studies. Please see lines 261-334 of the revised manuscript.

3) Line 244 – 245: The statement that hospital-acquired infections such as C. difficile "depict drug toxicity" is inaccurate. C. difficile is a complication of antibiotic use due to microbiome disruption, not a direct toxic effect. This should be rephrased. Furthermore, please describe the method used to confirm causality of all adverse drug reactions and provide data on the number and type of ADRs identified, including in which patient demographics they occurred.

Response: We have added the following clarifications in the text,

---

## [Decision Letter · Decision Letter 1]

16 Nov 2025

Factors influencing drug-susceptible tuberculosis treatment outcomes in Romania and Ukraine

PONE-D-25-40612R1

Dear Dr. Akkerman,

We’re pleased to inform you that your manuscript has been judged scientifically suitable for publication and will be formally accepted for publication once it meets all outstanding technical requirements.

Kind regards,

Frederick Quinn

Academic Editor

PLOS ONE

Additional Editor Comments (optional):

Reviewers' comments:

Reviewer's Responses to Questions

**Comments to the Author**

Reviewer #1: All comments have been addressed

2. Is the manuscript technically sound, and do the data support the conclusions?

Reviewer #1: Yes

3. Has the statistical analysis been performed appropriately and rigorously?

Reviewer #1: Yes

4. Have the authors made all data underlying the findings in their manuscript fully available?

Reviewer #1: Yes

5. Is the manuscript presented in an intelligible fashion and written in standard English?

Reviewer #1: Yes

Reviewer #1: (No Response)

**Do you want your identity to be public for this peer review?** For information about this choice, including consent withdrawal, please see our Privacy Policy

Reviewer #1: No

---

## [Editor Report · Acceptance letter]

PONE-D-25-40612R1

PLOS ONE

Dear Dr. Akkerman,

I'm pleased to inform you that your manuscript has been deemed suitable for publication in PLOS ONE. Congratulations! Your manuscript is now being handed over to our production team.

Kind regards,

on behalf of

Dr. Frederick Quinn

Academic Editor

PLOS ONE